# The Synthesis, Characterization and Anti-Tumor Activity of a Cu-MOF Based on Flavone-6,2′-dicarboxylic Acid

**DOI:** 10.3390/molecules28010129

**Published:** 2022-12-23

**Authors:** Jie Zhang, Tingting Jiang, Xinyu Song, Qing Li, Yang Liu, Yanhua Wang, Xiaoyan Chi, Jie Sun, Liangliang Zhang

**Affiliations:** 1School of Life Science, Ludong University, Yantai 264025, China; 2Ningbo Institute of Northwestern Polytechnical University, Northwestern Polytechnical University (NPU), Ningbo 315103, China

**Keywords:** copper, flavone-6,2′-dicarboxylic acid, crystal structure, anti-tumor activity

## Abstract

A novel two-dimensional copper(II) framework (LDU-1), formulated as {[Cu_2_(L)_2_·2NMP}n (H_2_L = flavone-6,2′-dicarboxylic acid, NMP = N-Methyl pyrrolidone), has been constructed under solvothermal conditions and characterized by single-crystal X-ray diffraction, infrared spectroscopy (IR), thermogravimetric analysis and powder X-ray diffraction (PXRD). In the crystal structure, the Cu(II) shows hex-coordinated with the classical Cu paddle-wheel coordination geometry, and the flavonoid ligand coordinates with the Cu(II) ion in a bidentate bridging mode. Of particular interest of LDU-1 is the presence of anti-tumor activity against three human cancer cell lines including lung adenocarcinoma(A549), Michigan cancer foundation-7 (MCF-7), erythroleukemia (K562) and murine melanoma B16F10, indicating synergistic enhancement effects between metal ions and organic linkers. A cell cycle assay indicates that LDU-1 induces cells to arrest at S phase obviously at a lower concentration.

## 1. Introduction

Flavonoids, which consist of two phenyl rings (A and B) and heterocyclic ring (C), are series of benzo-g-pyrone derivatives distributed ubiquitously in plants (Figure 1) [1,2,3,4]. In the past two decades, flavonoids have been reported to possess wide-ranging pharmacological properties such as anti-inflammatory [5], decreasing the expansion of the coronary artery [6], treating leukemia [7], inhibiting histamine-induced gastric acid secretion [8,9] and antiviral activity including coronavirus infection [10,11], and many other applications [12,13]. Flavonoids have also been suggested as a possible cancer chemopreventive agent on the basis of its inhibitory effects on tumor initiation, promotion, and progression [14,15,16].

So far, most of the reported research is mainly focus on the design and synthesis of flavone with hydroxyl and alkoxy groups and their biological activities. In contrast, only few works focus on flavone with carboxyl group. For example, Cutler et al. synthesized a series of substituted flavone-8-carboxylic acids, some of which showed solid tumor inhibition effect in mice bearing B16F10 melanomas [17]. Additionally, then, Zwaagstra et al. reported a series of flavone-6-carboxylic acid and flavone-8-carboxylic acid compounds through the chalcone route and investigated their inhibitory effects towards (leukotriene)D4 as effective anti-asthmatic agents [18,19]. In addition, Jurg et al. reported twenty-five flavone-6-carboxylic acid compounds and tested their ability to inhibit histamine-induced gastric acid secretion in rats [9]. Recently, Wen’s group prepared a series of flavone-6-carboxylic acid compounds through the chalcone route and studied their antibacterial activities [20].

Despite possessing several positive biological activities, flavonoids have seldom been used as drugs independently because of their poor solubility and bioavailability [21]. Many flavonoids are natural chelators, and flavonoid–metal complexes have shown significantly higher bioactivity compared with their parent flavonoids [22,23]. Roy et al. revealed the chemotherapeutic effects of vanadium luteolin complex against HT-29 human colon carcinoma cell line and the molecular mechanisms leading to induction of apoptosis and inhibition in the cell proliferation [24]. Ikeda et al. synthesized rutin-zinc(II), a flavonoid–metal complex, which showed more efficient antioxidant activity than free rutin and cytotoxicity against cancer cell lines in vitro and synergistic antitumor activity preventing side effects of chemotherapy [25]. Spoerlein et al. compared the (iso-)flavonoids chrysin, apigenin, genistein and their homoleptic copper(II) complexes for general cancer cell growth inhibition and for antimetastatic effects on rapidly proliferating and metastasizing 518A2 melanoma cells. The complexes were three to five times more active than the free flavonoids in cytotoxicity assays [26]. However, somewhat surprisingly, the coordination chemistry of flavonoid carboxylic acids is still an area that has not been paid enough attention to when the development of coordination chemistry of carboxylic acids has been very rapid and is still in full swing [27].

Copper is a mineral nutrient and participates in a variety of biological progresses including respiration, metabolism, cell signaling and so on [28]. Copper-based MOFs have excellent physicochemical properties among MOFs and have recently attracted wide attention in various biomedical applications [28]. Some nanoMOFs based on Cu^2+^, exert antiproliferative effect by transforming H_2_O_2_, an important compound in reactive oxygen species family that has higher concentrations in tumor tissues than in normal tissues, into highly cytotoxic ·OH through Fenton/Fenton-like reactions or catalytic effect [29]. Considering the pharmacological properties of flavonoids and the metal-assisted cancer therapy of copper, we focus on synthesizing some carboxyl-containing flavonoids and their complexes to explore their potential applications as new chemotherapeutic agents. In this work, we introduce carboxyl group into flavone through the chalcone route, successfully obtaining flavone-6,2′-dicarboxylic acid (H_2_L, Figure 2). Versatile reactions by varying the reagent ratios, reaction solvents, and other conditions have been systematically explored. Finally, the reaction of Cu(NO_3_)_2_, H_2_L with N-Methyl pyrrolidone (NMP), N,N-dimethylacetamide (DMA) and H_2_O in a 1:1:1 volume ratio at 80 °C gave the well-formed green cubic crystals of LDU-1. The in vitro antitumor activity investigations show that LDU-1 exhibits inhibitory activity against tumor cell A549, MCF-7, K562 and B16F10.

## 2. Results

### 2.1. Structural Description

LDU-1 crystallizes in the triclinic space group *P-1*, and the asymmetric unit consists of two copper ions, two different fully deprotonated L ligands, and two coordinated NMP molecules (Figure 1). Each metal ion is penta-coordinated with the classical Cu paddle-wheel coordination geometry, which is chelated by four H_2_L ligand and the axial coordination sites are occupied by two NMP molecules. Both carboxylic groups of H_2_L ligand were deprotonated and adopt a bidentate bridging mode to link two Cu ions. The Cu-O bond length range from 1.956(4) to 2.098(3) Å, in accordance with the ones of other CuII complexes [30,31]. The dihedral angles between ring B and C are 74.0 (18)° and 67.1 (84)°, respectively, showing that rings B and C are severely twisted across the C–C single bond, which provide the potentiality of the formation of a high dimensional structure. Furthermore, the H_2_L ligands link Cu paddle-wheel to form 2D layer structure (Figure 2). Table 1 contains the crystallographic details of LDU-1 and Table 2 collects the selected bond lengths and angles for LDU-1.

### 2.2. Morphology, Stability, Phase Purity and FT-IR Spectrum

The morphology and size of LDU-1 certified by SEM images demonstrates that it is nanoparticles with a size range of 10–40 nm (Figure 3a). The experimental powder PXRD pattern is almost identical to the corresponding simulated one, indicating the phase purity. In order to verify the stability of LDU-1 in the cell culture medium, we immersed it in the cell culture medium for 48 h, and the PXRD pattern showed no change compared with the as-synthesized one (Figure 3b). FT-IR spectrum of LDU-1 was also investigated. The sharp bands at about 1614 cm^−1^ and 1399 cm^−1^ are attributed to asymmetric and symmetric stretching vibrations of carboxylic group, respectively (Appendix A). To investigate the thermal stability of LDU-1, thermogravimetric analysis (TGA) of LDU-1 was performed under N_2_ atmosphere (Appendix A). LDU-1 has two identifiable weight loss steps: the first one is consistent with the removal of two coordinated NMP molecules (obsd 22.89%, calcd 21.05%), which appears between 215 and 285 °C. The second one is attributed to the collapse of the framework due to the removal of NMP, which is in the range of 286 to 390 °C.

### 2.3. In Vitro Antitumor Activity

The excellent thermal stability of LDU-1 and its stability in cell culture medium promote us to investigate the antitumor activity of it. The in vitro cytotoxic activities of LDU-1 and H_2_L against human lung adenocarcinoma cells A549, human breast cancer cells MCF-7, human erythroleukemia cells K562 and murine melanoma B16F10, were evaluated with CCK-8 assay, with cisplatin as positive control. As shown in Table 3, the anticancer activity of the ligand was relatively poor, but LDU-1 showed moderate inhibitory effects against A549, MCF-7, K562 and B16F10 compared with the antitumor activities of cisplatin, and IC_50_ values were within the range of 14.53–32.47 μM. This could be attributed to synergistic enhancement effects between metal ions and organic linkers. Among four cancer cells, LDU-1 was more sensitive to the suspended K562 erythroleukemia cell than other three adherent cells, which may be due to the greater number of contact areas between suspended tumor cell and LDU-1 in comparison with other adherent tumor cells, while all the tumor cells were more sensitive to LDU-1 than the normal cell NIH3T3. As shown in Figure 4, LDU-1 exhibited a dose-dependent cytotoxicity toward cells. When the concentrations of LDU-1 were 20 and 40 μM, the cell survival rates were 34.19% and 12.47% for K562 tumor cell, and 82.67% and 74.66% for NIH3T3 normal cell, respectively, indicating that LDU-1 had a more remarkable inhibitory effects toward K562 tumor cell compared with the normal cell NIH3T3 and that LDU-1 would be an effective antitumor drug with good selectivity.

To further investigate the effect of LDU-1 on cells, cell cycle assay was performed by flow cytometry with PBS as negative control and cisplatin as positive control. These results revealed that the proportion of tumor cells in the S phase increased (*p* < 0.05, Table 4), accompanied by a decreased proportion in the G1 phase and G2/M phase. The results indicated that LDU-1 induced the cells S phase arrest obviously at lower concentration (10 μM). In addition, LDU-1 could effectively affect the cell cycle compared with H_2_L and cisplatin (Appendix A).

## 3. Discussion

In this study, flavone-6,2′-dicarboxylic acid (H_2_L) was synthesized through the chalcone route. The copper complex (LDU-1) was obtained by the reaction of Cu(NO_3_)_2_ with H_2_L. LDU-1 showed moderate inhibitory effects against K562, A549, MCF-7 and B16F10, indicating a synergistic enhancement effects between metal ions and organic linkers. LDU-1 was more sensitive to the suspended K562 erythroleukemia cell than other three adherent cells, while all the tumor cells were more sensitive to LDU-1 than the normal cell NIH3T3, which manifested that LDU-1 would be an effective antitumor drug with good selectivity. Cell cycle assay indicated that LDU-1 induced cells to arrest at S phase obviously at a lower concentration.

## 4. Materials and Methods

### 4.1. Materials and Methods

3-Acetyl-4-hydroxybenzoic acid was prepared according to the literature [20]. Other chemicals were of reagent grade and used as commercially obtained without further purification. Elemental analyses (C, H or N) were carried out on a Perkin-Elmer 240 elemental analyzer. Powder X-ray diffraction measurements were performed with a Bruker AXS D8 Advance instrument. SEM image was recorded on a Hitachi scanning electron microscope Regulus 8100. The FT-IR spectra were recorded in the range of 4000~400 cm^−1^ on a Nicolet 330 FTIR Spectrometer using the KBr pellet method. Thermogravimetric analysis (TGA) experiments were performed using a Perkin-Elmer TGA 7 instrument (heating rate of 10 °C·min^−1^, nitrogen stream). NMR spectra were obtained on a Bruker Advanced 500 MHz spectrometer. The absorbance was determined on a Thermo Scientific, Multiskan MK3 microplate reader. Cell cycle data were obtained on a BD FACSCalibur flow cytometer.

### 4.2. Synthesis of Flavone-6,2′-dicarboxylic Acid (H_2_L)

3-Acetyl-4-hydroxybenzoic acid(**1**) was conveniently prepared in our laboratory via Fries rearrangement synthesis method from ethyl -4-acetoxybenzoate followed by hydrolysis and acidification. To an ice-cooled solution of **1** (5.0 g, 27.8 mmol) and 2-carboxybenzaldehyde (4.5 g, 30.0 mmol) in ethanol (75 mL) was added 40% KOH (25 mL). The resulting dark red solution was stirred until reaction was complete as judged by TLC. The reaction mixture was acidified to pH 2 with dilute hydrochloric acid, and a yellow precipitate formed. The yellow precipitate was filtered off, washed with H_2_O and dried. Finally, 2-hydroxychalcone-2,5′-carboxylic acid (**2**) was obtained by recrystallization of the above product in ethanol. The compound was dried in vacuum with a yield of 63%.

^1^H NMR (500 MHz, DMSO-*d*_6_): 12.89 (s, 1H, COOH), 11.87 (s, 1H, COOH), 8.36 (d, J = 2.05 Hz, 1H), 8.03 (dd, J = 8.7 Hz, 2.3 Hz, 1H), 7.86 (d, J = 7.65 Hz, 1H), 7.78(m, 2H), 7.62(t, J = 7.4 Hz, 1H), 7.10(d, J = 8.7 Hz, 1H), 6.13(s, 1H, OH), 3.73–3.94 (m, 2H) ppm (Appendix A).

**2** (3.12 g, 10 mmol) and I_2_ (0.05g, 0.2 mmol) were dissolved in 30 mL DMSO and heated to reflux with a microwave chemical reactor for 7 min at a power of 300 watts. After cooling to room temperature, the mixture was poured into 100 mL of ice water. The resulting white precipitate was filtered off, washed with saturated aqueous NaHSO_3_, and then with H_2_O. H_2_L was obtained after the above product was dried in vacuum at 80 °C for 6 h with the yield of 90%. The EIMS exhibited a molecular ion at m/z 311 [M+H]^+^ (Appendix A). ^1^H NMR (500 MHz, DMSO-*d*_6_): 13.34 (s, 2H, COOH), 8.63 (d, J = 2 Hz, 1H), 8.31 (dd, J = 8.75 Hz, 2 Hz, 1H), 7.94 (dd, J = 6.85 Hz, 1.85 Hz, 1H), 7.67–7.80 (m, 4H), 6.68 (s, 1H) ppm (Appendix A). ^13^C NMR (125 MHz, DMSO-*d*_6_): 176.66, 169.97, 166.15, 166.04, 158.14, 134.50, 132.41, 131.96, 131.26, 131.11, 130.19, 129.82, 128.25, 126.61, 122.92, 118.99, 110.3 ppm (Appendix A).

### 4.3. Synthesis of LDU-1

H_2_L (2 mg, 6.45 mmol) and Cu(NO_3_)2∙5H_2_O(1.8 mg, 6.45 mmol) were dissolved in NMP/DMA/H_2_O (1/1/1, *v*/*v*/*v*, 1mL). The solution was sealed in a glass tube, slowly heated to 80 °C from room temperature in 300 min, kept at 80 °C for 3000 min, and then slowly cooled to 80 °C in 600 min. The green cubic crystals that formed were collected, washed with DMA, and dried in air (Yield 35%). Anal. Calcd. for C_44_H_34_Cu_2_N_2_O_14_ (941.81): C, 56.65; H, 3.03; N, 3.00%. Found: C, 59.70; H, 3.05; N, 3.02%.

### 4.4. X-ray Crystallography

Single-crystal structure analysis of LDU-1 was performed at 293(2) K on an Agilent SuperNova diffractometer equipped with a copper micro-focus X-ray sources (λ = 1.54178 Å). The data were collected with an ω-scan mode in an arbitrary φ-angle. Data reduction was performed with the CrysAlisPro package, and an analytical absorption correction was performed. The structure was solved by direct methods and refined by full-matrix least-squares on *F*^2^ with anisotropic displacement using the SHELXL software package [32]. The non-H atoms were treated anisotropically, whereas the aromatic and hydroxyl hydrogen atoms were placed in calculated ideal positions and refined as riding on their respective carbon or oxygen atoms. The structure was examined using the Addsym subroutine of PLATON to ensure that no additional symmetry could be applied to the models [33].

### 4.5. Test for Antitumor Activity

LDU-1 (100 mg) was ground with a planetary ball mill in a 20 mL steel vessel with a 10 mm steel ball at 50 Hz for 10 min. The obtained powder was dispersed in DMSO and further diluted to different concentrations by culture medium. The final concentration of DMSO should be controlled in less than 0.01%. All cell lines, including human non-small cell lung cancer line A549, breast cancer cell line MCF-7, murine B16F10 melanoma cells, and human erythroleukemia cell line K562 were cultured in RPMI 1640 medium, and NIH3T3 were cultured in DMEM, supplemented with 5% fetal bovine serum (FBS) under 5% CO_2_ and 95% air at 37 °C. The cell counting Kit-8 (CCK-8) assay was performed to investigate the cytotoxicity. Briefly, cells were seeded into 96-well plates at 5 × 10^3^ cells per well and then exposed to the tested compound in DMSO at different concentrations (5, 20, 40, 80, and 160 μM) for 24h, 48 h, respectively. After treatment, 10 μL of CCK-8 was added to each well, and the plates were incubated for an additional 2 h. The absorbance of each well was detected by automatic microplate reader at 450 nm. The inhibition rate was calculated according to the absorbance value, and the IC_50_ value of the compound was calculated according to the inhibition rate. The experiments were performed in triplicate. Cisplatin, a commonly approved agent for the treatment of many tumors, was used as the positive control.

Cell cycle analysis was performed to further investigate the effect of LDU-1 on tumor cells. Briefly, A549 cells (1 × 10^6^) were incubated with or without various compounds for 8 h. Cells were collected, rinsed with cold PBS, and fixed in pre-cooling 70% ethanol overnight. Samples were washed twice with cold PBS and resuspended in RNase A (100 μg/mL) for 30 min. Then, 200 μL PI (50 μg/mL) was added to the samples at 4 °C for 30 min. Data from 100,000 events/sample were collected by the FACS and analyzed using FlowJo software.

## Data Availability

CCDC 1571044 contains the supplementary crystallographic data of LDU-1 for this paper. These data could be obtained free of charge via www.ccdc.cam.ac.uk/conts/retrieving.html (accessed on 30 August 2022) (or from the CCDC, 12 Union Road, Cambridge CB2 1EZ, UK; fax: +44-1223-336033; E-mail: deposit@ccdc.cam.ac.uk).

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
