# Peer review of "The Synthesis, Characterization and Anti-Tumor Activity of a Cu-MOF Based on Flavone-6,2′-dicarboxylic Acid"

_molecules, 2022, doi:10.3390/molecules28010129_

Round 1
Reviewer 1 Report
The authors reported the synthesis, characterization, and anticancer effect of a copper-based MOF based on a new flavone-based ligand. Flavonoids are a type important natural small-molecular bioactive compounds and their combination with metal ions could hold a great potential in various biological fields. Therefore, the study is of great significance. The manuscript is well-organized and written. However, some minor revisions are still needed.
1. Line 139, there is no chemical stability results of the LDU-1.
2. The mass spectrum characterization of H2L is missing.
3. Table 3, IC50 of Cu(NO3)2 should be studied to support the synergistic anticancer conclusion.
4. What is the size of LDU-1, which is very important for the anticancer effect. SEM study is needed.
5. Statistical analysis is needed for the results in Figure 3.
6. Some relevant literatures should be cited. For example, Adv. Colloid Interface Sci. 2022, 305, 102686. Acta Biomaterialia 2022, 152, 495-506.
Author Response
Please find our reply in the attachment.

Reviewer 2 Report
The current work titled " The synthesis, characterization and anti-tumor activity of a Cu-MOF based on flavone-6,2’-dicarboxylic acid " reported by Jie Sun and co-workers report A novel two dimensional copper(II) framework {[Cu2(L)2·2NMP}n with anti-tumor activity. The complex was characterized by single crystal X-ray diffraction, PXRD analysis, TGA and IR. However, a minor revision is required before it can be accepted for publication and I have the following concerns regarding the present work.
1、 The authors should pay more attentions to format of this manuscript. There are some grammar and spell mistakes.
2、 Is the complex stable in the cell experimental medium? Please add PXRD analysis of it soaked in cell culture solution for 48h.
Author Response
- We have corrected the grammar and spelling mistakes in the revised version.
- We have provided the PXRD pattern of LDU-1 after immersing it in the cell culture medium for 48h.